# Sex Differences and the Role of Gaming Experience in Spatial Cognition Performance in Primary School Children: An Exploratory Study

**DOI:** 10.3390/brainsci11070886

**Published:** 2021-07-01

**Authors:** Claudia van Dun, Alex van Kraaij, Joost Wegman, Jorrit Kuipers, Esther Aarts, Gabriele Janzen

**Affiliations:** 1Donders Institute for Cognitive Neuroimaging, Radboud University Nijmegen, 6525 XZ Nijmegen, The Netherlands; alex1vande5kraaijtjes@gmail.com (A.v.K.); joostwegman@gmail.com (J.W.); e.aarts@donders.ru.nl (E.A.); 2Green Dino, 6708 PB Wageningen, The Netherlands; jorrit@greendino.nl; 3Behavioural Science Institute, Radboud University Nijmegen, 6525 XZ Nijmegen, The Netherlands

**Keywords:** spatial cognitive development, object location memory, navigation, virtual reality, sex differences, gaming experience

## Abstract

Sex differences are repeatedly observed in spatial cognition tasks. However, the role of environmental factors such as gaming experience remains unclear. In this exploratory study, navigation and object-relocation were combined in a naturalistic virtual reality-based spatial task. The sample consisted of *n* = 53 Dutch children aged 9–11 years. Overall, girls (*n* = 24) and boys (*n* = 29) performed equally accurately, although there was an increase in accuracy with age for boys (*η_p_*^2^ = 0.09). Boys navigated faster than girls (*η_p_*^2^ = 0.29), and this difference increased with age (*η_p_*^2^ = 0.07). More gaming experience in boys versus girls (Cohen’s *d* = 0.88) did not explain any result observed. We encourage future confirmatory studies to use the paradigm presented here to investigate the current results in a larger sample. These findings could be beneficial for optimizing spatial cognition training interventions.

## 1. Introduction

Sex differences in cognitive abilities are repeatedly observed in the field of spatial cognition, often favouring males (see [1,2] for reviews). Spatial cognition can be described as the knowledge and cognitive representation of the structure, entities, and relations of space [3]. Spatial abilities are paramount to human survival for a wide variety of daily tasks, allowing us to use tools, estimate magnitudes, and navigate, among other examples. The origin of the difference in spatial abilities between the sexes is highly debated. Biological factors play a role, but so do environmental factors, and these factors further interact in a complex fashion [1,4]. In the exploratory study presented here, we examine differences in spatial cognition between boys and girls, taking into account both age and experience with playing computer games.

Spatial skills are consistently found to be malleable [1,5,6,7]. In a meta-analysis of 217 training studies, it was concluded that spatial skills could not only be successfully trained (weighted effect size Hedges *g* = 0.47), but also that the effects of training were stable and transferrable [6]. These results are broadly in line with an earlier meta-analysis on the same topic [7]. Given the trainability of spatial skills, it follows that sex differences could—at least in part—be due to more training in boys throughout childhood. One opportunity for training is through playing computer games. In a study including a large sample (*N* = 1278) of students, substantial sex differences in gaming experience favouring male students were found [8]. By investigating a smaller subset of the sample (*n* = 180), Terlecki and Newcombe (2005) moreover demonstrated that male students obtained higher scores than female students on a spatial task (Mental Rotation Task), and that this effect was mediated by gaming experience. In addition, *within* female students, those with more gaming experience performed better on the spatial task than those with less gaming experience. Another study assessed sex differences in spatial skills related to gaming experience on a different type of task, namely in spatial navigation tasks [9]. In line with Terlecki and Newcombe (2005), Richardson, Powers, and Bousquet (2011) concluded that the magnitude of the observed advantage in spatial skills in men over women is potentially influenced by exposure to video games [9]. Albeit merely correlational in nature, the findings from these studies do demonstrate that the sex difference in gaming experience is indeed associated with the sex difference in spatial skills.

Researchers have used a wide variety of tasks to assess spatial skills, including mental rotation, object location, and navigation tasks. Typical object location tasks include tasks such as the well-known game Memory, tasks in which participants have to remember a spatial sequence, and tasks in which participants are required to learn the locations of objects and reproduce their locations from memory after a certain delay. Opposed to other aspects of spatial cognition, such as mental rotation, where males are consistently found to outperform females across the lifespan [1], the investigation of sex differences in object location memory has yielded mixed results, with most studies finding a female advantage for object location memory [10,11,12], and some other studies finding a male advantage [13] or no sex difference at all [14].

Typical navigation tasks range from wayfinding and route learning tasks to free navigation tasks. Navigation requires spatial orientation through a sequence of spatial cues to be stored in order to find and remember the path. Male subjects typically perform better on navigation tasks than female subjects [15,16,17,18], although some studies found no sex differences or a female advantage under specific conditions, for example on tasks that rely heavily on verbal instructions (see [19] for a recent meta-analysis). The sex difference in navigation ability is often attributed to a sex difference in spatial strategy preference. Siegel and White (1975) proposed a model including three spatial strategies for spatial development and learning, depending on the kind of information the subject selects to navigate: (1) landmark, (2) route, or (3) survey strategies [20]. The landmark strategy uses perceptually salient or subjectively important cues to the subject for navigation. The route strategy is based on the routes connecting the landmarks. The landmark and route strategy both rely on egocentric reference frames, which are subject-centred. Contrarily, the survey strategy is based on a cognitive map of the environment using relations and distances between cues. The latter is viewed as the more sophisticated strategy, as it relies on allocentric reference frames based on the interrelations of available cues, irrespective of the subject’s own position [21,22]. Research has shown that female subjects prefer a landmark strategy, using the sequence of turns and proximal cues, whereas male subjects prefer a survey strategy using cardinal directions, metric estimations, and distal cues [14], and that the latter strategy is most beneficial for performance [23]. Tasks may vary in the degree to which certain cues are available, which in turn could affect the results and the likelihood of observing a sex difference.

Here, we use an object-relocation task in an open field within a virtual 3D city to assess memory for object location during free navigation. Virtual reality-based tasks have been applied successfully to test spatial skills both in adults [24] and in children [25]. In some studies, virtual reality tasks were found to be more suitable than traditional spatial tasks in detecting subtle differences in performance [24]. The current task encompasses, on the one hand, aspects of typical object location memory tasks and, on the other hand, aspects of typical navigation tasks. Both proximal and distal cues are available in the task to allow participants to use the navigational strategy they naturally prefer.

From a developmental perspective, in addition to the use of egocentric reference frames, children display the ability to use allocentric reference frames from around three years of age [26,27], although optimal integration of the two is not achieved until much later [28]. The development of spatial strategies continues throughout childhood and adolescence, and is dependent on experience navigating and exploring environments [29]. Between 7–12 years of age, children learn to integrate a larger amount of cues, including distal landmarks to infer object location [30]. Although it was once believed that sex differences in spatial cognition do not emerge until puberty, later work has shown sex differences to be present throughout childhood [31]. However, effect sizes are relatively smaller in children under thirteen compared with older children or adults [2,19,32], and depend on the spatial domain tested and the specific task characteristics of the tests being used (for a recent review, see [33]). Given the important developmental processes for spatial abilities taking place in late childhood, we here include children from 9–11 years of age and assess how age relates to spatial cognitive performance at this age.

In summary, the investigation of sex differences on tasks involving object location and navigation has yielded contradictory findings, and little is known about sex differences on these specific tasks in children. Additionally, the specific role of gaming experience remains unclear. Therefore, the research question of the current study is: What are the differences in spatial cognition between boys and girls of 9–11 years of age, and what is the role of gaming experience? Other studies investigating spatial abilities in children pointed out the importance of using tasks that are specifically designed to test children, as opposed to adults [34,35]. Here, we use a preliminary version of a professionally designed computer game for children (GreenDino, Wageningen, The Netherlands) to explore the effects of sex, age, and gaming experience in this new paradigm. The game takes place in a virtual city, including both proximal and distal cues, as would be available in a real-life city. In addition, the game incorporates a first-person perspective. Together, this creates a naturalistic environment for navigation and testing spatial abilities. We use various starting positions in the testing phase (i.e., identical versus different from the encoding phase) to ensure a variety in task difficulty and to allow usage of different spatial strategies. We assess performance by not only taking accuracy as an outcome measure, but also navigational speed, and the degree to which participants circumnavigate in the testing phase. Based on the literature described above, we expect to find a sex difference in performance and task performance to improve with gaming experience and age.

## 2. Materials and Methods

### 2.1. Participants

The final sample for this study consisted of 53 Dutch primary school children (24 girls, 29 boys, *M*_age_ = 10.06, *SD*_age_ = 0.74, and Range_age_ = 9–11 years old; see Table 1 for additional demographic information of the participants). A total of 119 children participated in the study. However, the parents of 53 children did not (fully) complete the survey, and 13 children were unable to complete the first level of the game within the given time. One child was excluded for not reaching the minimum of 3 trials per condition after filtering (see data preparation section below). We recruited the children via their primary schools participating in a project of the Science Education Hub of the Radboud University. The first group of children (*n* = 75/*n* = 35 for final sample) was tested in April 2017 at the Donders Institute in Nijmegen, when their school visited the research facility. The second group of children (*n* = 44/*n* = 18 for final sample) was tested in May 2017, when researchers of the Donders Institute visited their primary school. As an indicator for the socioeconomic status of households of the children tested in the final sample, the education level of the mother was low in 13.2 percent, middle in 26.4, and high in 60.3 percent, and the education level of the father was low in 7.5 percent, middle in 45.3 percent, and high in 47.2 percent. For comparison, the overall average education level in 2018 in the Netherlands was 29 percent low, 40 percent middle, and 30 percent high [36]. Children participated voluntarily in the study, and did not receive any form of compensation. We obtained written informed consent from both parents of all participating children. The local ethical committee of Social Sciences of the Radboud University Nijmegen approved the study.

### 2.2. Materials

#### 2.2.1. Questionnaire

Parents of the participating children filled out an online questionnaire about their child from their home computer. In addition to questions about daily food intake (not presented here), the questionnaire included questions about the child’s sex, age, and experience with playing computer games. Three questions were used to calculate gaming experience scores. The first question, “How often does your child play computer games?” could be answered on a 5-point Likert scale ranging from “Once per month or less” to “Multiple times per day”. The second question was: “In the past six months, how many hours per week did your child play computer games on average?” and was quantitative. The third question, “In comparison with children of the same age, how experienced is your child with computer games?” could be answered on a 6-point Likert scale ranging from “Very unexperienced” to “Very experienced”. For data analyses, we combined the answers to the three questions by first calculating z-scores and then taking the average.

Additionally, we asked parents to fill out the highest education level of either of the parents. Scores ranged from 1 to 5, indicating primary school level to university level, respectively. We used the maximum of these two scores to control for parental education level in our analyses.

#### 2.2.2. Spatial Cognition Game

In the current study, we used an early version of a game that was later used in a neuroimaging study to assess spatial cognition in children (recordings from the game can be found here: https://youtu.be/FyFKNf6F95o, accessed on 9 March 2021). The game was specifically designed for this purpose in collaboration with a virtual reality company (Green Dino, Wageningen, The Netherlands). The task was created and administered in Unity software. The first group of children played the game in one of the computer rooms of the Donders Institute on a 24-inch computer screen with a resolution of 1920 × 1080 pixels (*w* × *h*). The second group of children played the game on a 15-inch laptop with a resolution of 1920 × 1080 pixels (*w* × *h*) at their own primary school in a separate classroom reserved for the current study. Testing took place simultaneously in groups of 8 to 30 children. In each group, children received standardized oral instructions that were read aloud by one of the experimenters. Children were instructed to refrain from talking for the duration of the experiment. One or more experimenters were present in the test room for the complete duration of the task. After completion of the task, children were thanked for participating and brought back to their classmates by one of the experimenters.

#### 2.2.3. Virtual Environment

The game took place in a virtual city, with a square-shaped park in the middle. The object used in the task was a puppy. To increase visibility from afar, balloons were attached to the puppy. There were eight entrances to the park: one on each corner and one on the middle of each side, corresponding to the main axes and the diagonals (see Figure 1, numbers 1–8). The entrances served as starting positions at the beginning of each trial (see below for a detailed description of the trial structure). Two types of navigational cues were present: (1) four large buildings (church, ferry wheel, mill, and skyscrapers) positioned at the end of the horizontal and vertical axes of the city served as distal cues (see Figure 1, letters A–D), and (2) two identical trees with distinctly coloured leaves positioned inside the park served as proximal cues (see Figure 1, bright green and orange dot in the middle). Subjects played the task on a computer with a keyboard and a mouse. They could use the mouse to look around, and they could use the arrow buttons to move forwards, backwards, to the left, and to the right, correspondingly.

#### 2.2.4. Trial Structure

Players entered each trial in a first-person perspective. Each trial consisted of three phases: (1) encoding, (2) testing, and (3) feedback (see Figure 2 for example screenshots of the game). For every trial, the object was positioned at a random location in the park. In the encoding phase, subjects navigated towards the target object and collected it with a button press. In the testing phase, subjects were placed back either at their original starting position matching the starting position in the encoding phase (level 1) or at one of the seven remaining starting positions, different from the starting position in the encoding phase (level 2). The goal was to relocate the object at its original location as fast and accurately as possible, by navigating to the estimated original location and pressing the space bar. A red dot with surrounding grey circles indicated the exact position where the object would be replaced if the space bar were pressed. In the feedback phase, the original location of the object was visually shown, together with the points obtained for that trial. The minimum amount of points per trial was 10 to maintain motivation. The maximum amount of points per trial was 100. Points obtained increased linearly with decreasing difference in distance in virtual meters between the original location of the object in the encoding phase and the relocated location of the object in the testing phase.

#### 2.2.5. Level Structure

In the first level, the starting position in the encoding and testing phase was always identical. Once subjects obtained 500 points, the game would proceed to the second level. In the second level, the starting position in the testing phase was always different from the encoding phase to stimulate the use of allocentric strategies. After obtaining 500 points in the second level, or after 45 min of the experiment had passed, the experiment was completed.

#### 2.2.6. Outcome Measures

We selected three outcome measures from the game to be used for subsequent analyses: (1) Object distance, (2) Speed, and (3) Detour. The first outcome measure, object distance, was defined as the distance error in virtual meters (vm) between the chosen object location in the testing phase and the original object location in the encoding phase. A higher object distance indicated less accurate or precise object relocation. We chose to use this variable as opposed to the scores obtained by the player in the game (see Section 2.2.4), as object distance reflects the actual distance in virtual meters, whereas the scores obtained in the game were a linear transformation of this object distance. The second outcome measure, speed, was defined as the vm walked by the player in the testing phase divided by the duration of the testing phase in seconds (s), resulting in vm/s. Higher scores on this variable represent faster navigational speed. The third, detour, was defined as the vm walked by the player minus the length of the shortest path between the player’s start and end position in the testing phase, resulting in a measure for additional vm walked compared to walking in a straight line to the chosen end position, which in turn could be interpreted as a measure for goal directed behaviour. Higher scores on this variable reflect more circumnavigation in the testing phase. These three outcome measures together indicate efficient task performance.

### 2.3. Data Analysis

#### 2.3.1. Data Preparation

Frame data of the spatial cognition game was collected at 50 Hz. The data frames were then combined into data per trial. We filtered out trials on which children navigated out of the park, trials on which children did not move at all during the testing phase (i.e., relocated the object at their starting position (≤1 vm) in the testing phase), and trials that had a duration of more than two standard deviations above the mean for either the encoding (*M* = 25.58 s, *SD* = 23.53 s) or the testing phase (*M* = 23.13 s, *SD* = 15.46 s). Thereafter, we calculated average object distance, speed, and detour per subject per start angle condition as described below.

The starting position in the testing phase was either identical to the starting position in the encoding phase (0 degrees shift) or the starting position was shifted by 45, 90, 135, or 180 degrees. For subsequent analysis, we contrasted no shift (0 degrees) versus any shift (either 45, 90, 135, or 180 degrees). Shifts could be either clockwise or counter-clockwise, but this was not included as a separate variable for analysis. We set a minimum amount of 3 completed trials per condition to be included for analysis. This led to the exclusion of one participant, who only completed one trial within two standard deviations from the mean in the ’no shift’ condition. After filtering, subjects completed 6 no shift trials and 8 shift trials on average, for which we then calculated mean object distance, speed, and detour to be used in our main analysis.

#### 2.3.2. Analyses

We first reported the descriptive statistics to characterize our sample. Then, we used a repeated-measures MANCOVA with ‘start angle’ (no shift, shift) as within-subject factor, ‘sex’ (boy, girl) as between subject factor, and standardized ‘gaming experience score’ and standardized ‘age’ in months as covariates to answer our research question. Standardized ’parental education’ was added to the model as a covariate of non-interest. The dependent variables of the analysis were object distance, speed, and detour.

We used Cohen’s conventions [37] to evaluate the magnitude of the effect sizes. Thus, a Cohen’s *d* of 0.2 is considered small, 0.5 medium, and 0.8 large; a correlation of 0.1 is considered small, 0.3 medium, and 0.5 large; and a (multivariate) partial eta-squared of 0.01 is considered small, 0.06 medium, and 0.14 large. Differences of *p* < 0.05 were considered significant and *p* = 0.05–0.1 marginally significant.

We used R Studio and R version 3.5.1 with the following packages to process and visualise the data: Tidyverse version 1.2.1, Lubridate version 1.7.4, GGally version 1.4.0, Jsonlite version 1.5, and Haven version 1.1.2. In addition, we used IBM SPSS Statistics version 23 for the statistical analyses.

## 3. Results

### 3.1. Descriptive Statistics

To characterize our sample and explore the data, we ran a series of independent *t*-tests (uncorrected) to compare girls and boys on several parameters. We observed no significant difference between girls and boys with respect to age. On average, girls had less gaming experience than boys (*t*(51) = −3.24, *p* = 0.002). This effect was large (Cohen’s *d* = 0.89). The parental education level was higher for boys than for girls (*t*(51) = −3.40, *p* = 0.001). This effect was large (Cohen’s *d* = 0.93). Therefore, we controlled for parental education in our subsequent analyses. See Table 1 for an overview.

For the game, results indicated that, on average, girls and boys completed an equal amount of trials in total (*t*(51) = 0.35, *p* = 0.73; *M*_girls_ = 15.75, *SD*_girls_ = 2.42, *M*_boys_ = 15.52, *SD*_boys_ = 2.46). However, on average girls completed level 1 (i.e., obtained 500 points) in 5.54 (*SD* = 1.02) trials, whereas boys completed level 1 in 6.14 (*SD* = 0.95) trials. This difference was significant (*t*(51) =−2.20, *p* = 0.03) and of medium size (Cohen’s *d* = 0.61). For level 2, the difference in the amount of trials completed was not significant (*t*(51) = 1.47, *p* = 0.15; *M*_girls_ = 8.67, *SD*_girls_ = 2.51, *M*_boys_ = 7.79, *SD*_boys_ = 1.8). In the encoding phase, the distance between the player’s start position and the object to be collected was the same for girls and boys (*t*(51) = −1.20, *p* = 0.24; *M*_girls_ = 26.92 vm, *SD*_girls_ = 1.88, *M*_boys_ = 27.51, *SD*_boys_ = 1.67), serving as a manipulation check to confirm that the randomization procedure to determine the object position worked correctly. Despite similar object distances over trials, girls required more time to complete the encoding phase (*t(51)* = 3.45, *p* = 0.001; *M*_girls_ = 21.31 s, *SD*_girls_ = 8.85, *M*_boys_ = 15.07, *SD*_boys_ = 3.74), indicating that boys were faster in the encoding phase than girls. This effect was large (Cohen’s *d* = 0.92). In the testing phase, girls and boys walked similar distances (*t*(51) = 1.14, *p* = 0.26; *M*_girls_ = 33.29 vm *SD*_girls_ = 5.09, *M*_boys_ = 31.67 vm, *SD*_boys_ = 5.22). On average, girls spent more total time playing the game, both in level 1 (*t*(51) = 2.39, *p* = 0.02; *M*_girls_ = 181.31 s *SD*_girls_ = 58.14, *M*_boys_ = 140.39, *SD*_boys_ = 65.06) and in level 2 (*t*(51) = 4.31, *p* < 0.001; *M*_girls_ = 228.10 s, *SD*_girls_ = 67.39, *M*_boys_ = 156.09, *SD*_boys_ = 54.35).

Subsequently, we performed a series of Pearson’s correlation analyses. There was no significant correlation between age and gaming experience scores (*r* = 0.05, *p* = 0.71). The correlations between the three dependent variables for the repeated-measures MANCOVA, object distance, speed, and detour, are presented in Table 2, as well as the corresponding descriptive statistics. For the main analysis presented below, we chose a multivariate approach, as the three dependent variables, object distance, speed, and detour are conceptually related within the framework of spatial cognition, and the three variables together indicate efficient task performance. In addition, object distance and detour were moderately anti-correlated (*r* = −0.36, *p* = 0.009), which has been shown to enhance power within the framework of multi-variate analyses of variance [38], such as the repeated-measured MANCOVA used below. In addition, the multi-variate approach is free of sphericity assumptions and limits the joint error rate, as opposed to running a series of univariate repeated-measure ANCOVAs [39].

Through four independent sample *t*-tests (uncorrected), we could confirm that none of the dependent variables were significantly different between the two groups of participants tested on different days at different locations (*p* = 0.12–0.75). Additionally, through a Chi-square test, we confirmed that the composition with respect to sex of the two groups tested on different days did not differ significantly (*Χ*^2^(1) = 1.16, *p* = 0.28). Therefore, ‘test day’ was not included as a between-subject factor in the subsequent main analyses.

### 3.2. Repeated-Measures MANCOVA on Object Distance, Speed, and Detour

After confirming that our data met the assumptions of normality, homogeneity, and linearity, we conducted a repeated-measures MANCOVA with ‘start angle’ (no shift, shift) as within-subject factor, ‘sex’ (boy, girl) as between subject factor, standardized ‘age in months’ and standardized ‘gaming experience’ score as covariates, and object distance, speed, and detour as dependent variables. We controlled for standardized maximum ‘parental education’ level by adding it as a covariate of non-interest. Alongside the listed main effects, we included the following within-subject two-way interaction effects in the model: ‘start angle × sex’, ‘start angle × age’, ‘start angle × gaming experience’, ‘start angle × parental education’; together with the following within subject three-way interaction effects: ‘start angle × sex × age’, and ‘start angle × sex × gaming experience’. In addition, we included the following between-subject two-way interaction effects in the model: ‘sex × gaming experience’, and ‘sex × age’. The results are presented below.

#### 3.2.1. Multivariate Tests

The multivariate effect of start angle was significant (multivariate *F*(3,44) = 22.27, *p* < 0.001) and large (multivariate *η_p_*^2^ = 0.60). The start angle × age interaction effect was also significant (multivariate *F*(3,44) = 2.84, *p* = 0.049) and large (multivariate *η_p_*^2^ = 0.16).

The multivariate tests for the main effects of sex and age were also significant (multivariate *F*(3,44) = 6.99, *p* = 0.001, and multivariate *F*(3,44) = 5.20, *p* = 0.004, respectively). Both effects were large (multivariate *η_p_*^2^ = 0.32, and multivariate *η_p_*^2^ = 0.26, respectively). The multivariate test for the interaction between sex × age was marginally significant (*F*(3,44) = 2.70, *p* = 0.057). This effect was large (*η_p_*^2^ = 0.16).

The multivariate effect of gaming experience was not significant (multivariate *F*(3,44) = 0.72, *p* = 0.55, multivariate *η_p_*^2^ = 0.05), and neither was the sex × gaming experience interaction (multivariate *F*(3,44) = 0.13, *p* = 0.95, multivariate *η_p_*^2^ = 0.008). The multivariate tests for the main effect of parental education and all remaining two- and three-way interaction effects included in the model were also not significant (*p* = 0.54–0.83). The corresponding univariate tests for these non-significant results will therefore not be further described below.

#### 3.2.2. Univariate Tests

The univariate test for start angle was significant for object distance (Figure 3B,C; *F*(1,46) = 60.72, *p* < 0.001), speed (Figure 4B,C; *F*(1,46) = 8.06, *p* = 0.007), and detour (Figure 5B,C; *F*(1,46) = 12.23, *p* = 0.001). All three effects were large (object distance *η_p_*^2^ = 0.57, speed *η_p_*^2^ = 0.15, and detour *η_p_*^2^ = 0.21). Estimated marginal means indicated that players were more accurate and circumnavigated less on the ‘no shift’ trials (object distance *M* = 2.99 vm, detour *M* = 3.77 vm), as compared to ‘shift’ trials (object distance *M* = 5.82 vm, detour *M* = 7.13 vm). However, players were also slower on ‘no shift’ (speed *M* = 1.63 vm/s) than on ‘shift’ trials (speed *M* = 1.78 vm/s).

The univariate start angle x age interaction effect was not significant for object distance (Figure 3B,C; *F*(1,46) = 0.06, *p* = 0.81, *η_p_*^2^ = 0.001), and marginally significant for both speed (Figure 4B,C; *F*(1,46) = 3.09, *p* = 0.09) and detour (Figure 5B,C; *F*(1,46) = 3.63, *p* = 0.06). The latter effects were both of medium size (*η_p_*^2^ = 0.06, and *η_p_*^2^ = 0.07, respectively). Post hoc regression analyses using the same design as in our main analysis but without start angle as a within-subject variable and using either speed on the ‘no shift’ trials, or speed on the ‘shift’ trials as a dependent variable instead, indicated that as children got older they became significantly faster on ‘no shift’ trials (*t*(46) = 2.88, *p* = 0.006, *b* = 0.19), but only marginally on ‘shift’ trials (*t*(46) = 1.74, *p* = 0.09, *b* = 0.12). Post hoc regression analyses for detour using the same approach, indicated that, as children got older, the amount of detours increased more on the ‘shift’ trials (*b* = 1.81) compared to the ‘no shift’ trials (*b* = 0.67). However, it should be noted that both individual *b*-weights were not significantly different from zero (’no shift’: *t*(46) = 0.79, *p* = 0.43; ‘shift’: *p* = 1.31 and *p* = 0.20).

For sex, the univariate test was not significant for object distance (Figure 3A; *F*(1,46) = 0.04, *p* = 0.85, *η_p_*^2^ = 0.001), or detour (Figure 5A; *F*(1,46) = 1.30, *p* = 0.26, *η_p_*^2^ = 0.03), The univariate test for speed (Figure 4A) was significant (*F*(1,46) = 18.70, *p* < 0.001). This effect was large (*η_p_*^2^ = 0.29). Based on the estimated marginal means, we observed that boys (*M* = 1.90 vm/s) were generally faster than girls (*M* = 1.50 vm/s). To further investigate the absence of a difference in object distance between girls and boys, we wanted to explore possible differences in performance across trials to investigate differential learning effects between girls and boys. To that end, we analysed the individual regression slopes of the object distance per subject and per level for the first five trials (i.e., the minimum amount of trials required to complete a level). We extracted the slopes through the method described by Pfister and colleagues [1], and subsequently performed a post hoc *t*-test comparing girls and boys on level 1 and level 2, separately. Results indicated that there were no differences between girls and boys in learning across trials regarding object distance for level 1 (*t*(51) = 0.19, *p* = 0.85; *M*_girls_ = −0.41, *SD*_girls_ = 1.21, *M*_boys_ = −0.34, *SD*_boys_ = 1.57), or for level 2 (*t*(51) = 0.20, *p* = 0.84; *M*_girls_ = −0.39, *SD*_girls_ = 1.30, *M*_boys_ = −0.31, *SD*_boys_ = 1.35). To further investigate the absence of a difference in detour between girls and boys, we followed the same approach as described above for object distance, to explore possible differences between girls and boys in learning across trials in detour. We again calculated the individual regression slopes [40], but now for detour per subject and per level for the first five trials (i.e., the minimum amount of trials required to complete a level), and subsequently performed another post hoc *t*-test to compare girls and boys on level 1 and level 2, separately. Results indicated that there were no differences between girls and boys in learning across trials with respect to detour for level 1 (*t*(51) = −0.90, *p* = 0.36; *M*_girls_ = −9.56, *SD*_girls_ = 22.19, *M*_boys_ = −4.81, *SD*_boys_ = 16.46), or for level 2 (*t*(51) = −1.16, *p* = 0.25; *M*_girls_ = −1.90, *SD*_girls_ = 4.67, *M*_boys_ = −0.49, *SD*_boys_ = 4.23). Finally, as a follow-up analysis, we explored possible speed-accuracy trade-offs on the trial level by testing the correlation between speed and object distance, separately for girls and boys, and separately for level 1 and 2. To this end, we first filtered the trial data by removing extreme outliers on both speed and object distance to exclude inattentive trials from the analysis (>3 *SD*s above or below *M*; filtering out 7 out of 745 trials in total). No significant correlations between speed and object distance were found on the trial level for girls or boys in level 1 or in level 2 (*p*-values = 0.48–0.65).

For age, both univariate tests for object distance (Figure 3A; *F*(1,46) = 11.74, *p* = 0.001), and for speed (Figure 4A; *F*(1,46) = 4.70, *p* = 0.04) were significant. The effect for object distance was large (*η_p_*^2^ = 0.20), and the effect for speed was medium (*η_p_*^2^ = 0.09). The parameter estimates indicated that object distance decreased with age (*b* = −0.53), whereas speed increased with age (*b* = 0.11). The univariate test for detour (Figure 4A) was not significant (*F*(1,46) = 2.75, *p* = 0.104, *η_p_*^2^ = 0.06).

The univariate test for the sex x age interaction was significant for object distance (Figure 3A; *F*(1,46) = 4.57, *p* = 0.04). This effect was medium to large (*η_p_*^2^ = 0.09). The sex x age interaction was marginally significant for speed (Figure 4A; *F*(1,46) = 3.52, *p* = 0.07). This effect was medium (*η_p_*^2^ = 0.07). For detour, the test was not significant (Figure 5A; *F*(1,46) = 0.10, *p* = 0.75, *η_p_*^2^ = 0.002). Post hoc regression analyses for boys and girls separately using age, gaming experience, and parental education as predictors, and average object distance over both start angle conditions as the dependent variable, revealed that boys become more accurate in replacing the object as they get older (*t*(25) = −3.64, *p* = 0.001, *b* = −0.81), whereas the accuracy of girls remains constant with age (*t*(20) = −1.04, *p* = 0.31, *b* = −0.21). A second set of post hoc regression analyses for boys and girls separately using the same predictors, but using average speed over both start angle conditions as the dependent variable, showed a similar pattern: boys become faster as they get older (*t*(25) = 2.18, *p* = 0.04, *b* = 0.15), whereas the speed of girls remains constant with age (*t*(20) = 0.41, *p* = 0.69, *b* = 0.02).

## 4. Discussion

Here, we investigated the relation between sex and spatial cognition in 9–11-year-old children, including the role of gaming experience and age. We adopted a virtual object-relocation task to assess our research question. We used three parameters for task performance: (1) object distance (i.e., inaccuracy), (2) navigational speed, and (3) detours (i.e., circumnavigation). Combined, these three parameters indicate efficient task performance. In general, we found that boys performed faster on the task, and their navigational speed increased with age. The same interaction between sex and age was found for object distance, albeit we observed no main effect of sex. To our surprise, we found no effect of gaming experience on any of the outcome measures. In addition, for both boys and girls, performance was generally better on the identical starting positions condition compared with the different starting positions condition, and this difference increased with age, albeit only for navigational speed. We discuss our findings in more detail below.

First, girls and boys were equally precise in replacing the object as close as possible to its original location (i.e., object distance). A post hoc test also revealed no learning differences in object distance across trials between boys and girls. Analysis of the interaction between sex and age on object distance revealed that boys perform more accurately as they get older, whereas the accuracy of girls remains constant with age. The general increase in accuracy with age was therefore due to the increase in performance of boys, not girls. Second, we found that boys generally navigated faster than girls. Navigational speed during the testing phase was used for the main analysis, however, navigational speed was also higher for boys in the encoding phase. The observed interaction between sex and age on speed indicated that boys become faster with age, whereas the navigational speed of girls remains constant with age. Thus, the general increase in speed with age observed is again explained by the increase in speed for boys alone. Third, girls and boys circumnavigated (i.e., detoured) to a similar extent whilst replacing the object. A post hoc test again indicated no learning differences in circumnavigation across trials between boys and girls. We found neither a change in circumnavigation with age, nor an interaction effect.

Our results with respect to sex and speed are broadly in line with a study assessing sex differences in spatial cognition in 8–10-year-old children using the Virtual Morris Water Maze [41]. In that study, boys performed faster than girls, and performance for boys increased over trials, whereas the performance of girls remained constant. Age was not considered in the analyses, so it remains unclear whether the sex difference was moderated by age. Boys in that study also showed superior ability to navigate to the target location, whereas in our study relocation of the object was equally accurate for boys and girls. Other studies, assessing both sex and age in relation to spatial cognition in children, typically include a wider age range. One study assessed spatial cognition in a virtual object location task in children from three age groups between 4 and 12 years old [34]. In that study, performance was better for the older age groups, and girls took longer to complete the task in all age groups, in line with our results. However, the interaction between sex and age was not tested directly in that study. It should be noted that only the youngest age group (5–6 years old) differed from the two older age groups (7–9 and 10–12 years old), whereas the two older groups did not differ from one another. Another study, assessing spatial cognition in five age groups between 4 and 10 years of age, found that boys displayed superior spatial abilities in the hardest version of the virtual reference memory navigation task in the 6 and 7–8-year-old group, but not in the 9–10-year-old group [25]. In our study, we find differences in performance with age within our smaller age range of 9–11 years of age. Methodological differences make these studies hard to compare; notwithstanding, our findings suggest that using age in months as a continuous variable may reveal sex and age effects that remain unnoticed when grouping ages together.

Boys navigated faster than girls, both in the encoding and in the testing phase, although both performed with equal accuracy, and equal amount of circumnavigation. One possible explanation for this difference in performance between boys and girls in our task could have been gaming experience. Analysis of the questionnaire revealed that boys indeed had more gaming experience than girls. This finding is in line with the findings from a large study that observed that male students had more gaming experience than female students [8]. Here, we show that this difference in gaming experience favouring males is already present in much younger children. Despite the observed sex difference, gaming experience did not affect any of the outcome measures in the current task. Gaming experience also did not affect the relation between sex, age, or the starting position condition and any of the outcome measures. Hence, the observed sex difference in performance in our study cannot be explained by a difference in gaming experience. Our findings are contradicted by results from the study in 5–12-year-old children using a virtual object location task discussed above [34]. In that study, they found that navigational speed increased with more gaming experience. Contrary to our desktop version of a spatial cognition game, their game was presented on a 3D stereoscopic screen and subjects interacted with the game using a gamepad. Moreover, Rodriguez-Andres and colleagues (2018) used a training phase to get the players acquainted with the interaction method. The interaction method used in our study combined both the mouse and the arrow buttons on the keyboard in a way that is less common in video games. Together with the lack of training, the novel interaction method may explain why players with more gaming experience could not benefit from their previous experience in the current task. This interpretation assumes that gaming experience does not influence spatial skills per se, but rather the perceptual and motor skills required to play a spatial cognition game, at least in children. However, in an experimental study in which young adults played a popular first-person puzzle-platform video game, spatial skills were shown to increase thereafter [42]. Generally, first-person video games have been found to positively affect spatial skills [43,44]. We operationalized gaming experience in our study based on typical questions found in the literature; however, we did not assess specifically the type of games habitually played by the children. Therefore, we cannot rule out the possibility that children did not engage in this specific type of video game, which in turn could explain why no effect of gaming experience was found. Future studies should also include questions on the specific type of gaming experience, especially as the preferred type of games played by children may be different from that of adults. One example of such a questionnaire is the Survey of Spatial Representation and Activities (SSRA) which includes not only items on frequency of video game playing and experience, and preferences for doing so, but also items on the genre and console choice [8]. Null effects are hard to interpret, but it could also be the case that gaming experience does not have an effect on the spatial skills of children in this age range, for example as mass exposure might be necessary for an effect to emerge [9]. Thus, gaming experience at this age may not yet be sufficient to yield an effect. Another possibility is that the effects of playing video games are different for children than for adults. However, in a recent review the authors concluded that training using video games benefits the spatial skills of both children and adults to a similar extent [6].

In the game, players would start the testing phase either from the same position as in the encoding phase, or from a different starting position. We found that players were more accurate and circumnavigated less on the trials with identical starting positions, compared to the trials with different starting positions. Players were also slower on the trials with identical starting positions as opposed to the trials with different starting positions. The latter can most likely be explained by practice effects, considering that the different starting positions trials were only included in the second level of the game. Nevertheless, despite being slower on the identical starting position trials, players were both more accurate and circumnavigated less. Therefore, we can conclude that performance was overall more efficient on these trials. Based on the interaction between the starting position conditions and age, we noticed that as children get older, the increase in navigational speed was larger on the identical trials than on the trials with different starting positions. We also observed that the amount of detours taken on the trials with different starting positions increased more with age than on the trials with identical starting positions. The difference in accuracy between the two conditions, favouring the identical starting positions, was not affected by age. Given the increase in speed and decrease in detours, we conclude that overall efficiency increases with age on the identical starting position trials, but not on the different starting position trials. Possible explanations for the lack of an increase in performance with age on the different starting positions condition are that these trials were potentially too confusing or too hard to observe age effects in a single measurement. For future studies, it would be interesting to investigate the course of performance on the two trial types with age over several sessions, using a mixture of the two trial types across levels.

Difficulty may not only affect the detection of age effects; it also seems to be a crucial factor for detecting sex differences. We found that none of the results with respect to the starting positions was affected by sex. Rodriguez-Andres and colleagues (2018) argued that the absence of a sex difference on the scores obtained in their spatial cognition task was due to limited task difficulty. This is in line with other authors suggesting that sex differences in spatial abilities require sufficient task difficulty to emerge [25]. In earlier research, however, authors argued that spatial tasks were often too difficult to identify sex differences in children [31]. Here, the two starting position conditions introduce a difference in difficulty, as identical starting positions would be easier than different starting positions, as demonstrated by increased object distance and decreased detours (but slower speed) in our data. With this difficulty manipulation, we would have expected different sex effects depending on starting position, which was not the case. One explanation for the lack of a sex effect could be that the difference in difficulty between the conditions was not sufficient to yield an effect. Yet, we do find differences in accuracy, speed, and circumnavigation between the two conditions, regardless of sex, indicating better overall performance on the identical starting position trials. We also find overall sex effects irrespective of starting position, indicating that the difficulty of the task was generally adequate to observe sex differences.

Alongside an arguable difference in difficulty, it is important to note that the two starting position conditions may affect spatial strategies differently. Identical starting positions allow for task performance through egocentric strategies. Hence, the position of the target could be located by re-walking the same path in the testing phase as was chosen in the encoding phase. Alternative start positions, however, stimulate allocentric strategies. In that condition, replacement of the target could be based on inferring its location through the relations between the target and the available cues in the environment, irrespective of the position of the subject. In some studies, it was found that girls prefer spatial strategies based on egocentric cue use, whereas boys prefer spatial strategies based on allocentric cue use [14]. Additionally, shifting the starting position requires mental rotation, an ability that is typically higher in males [1]. Therefore, we expected that girls would be more affected by the shift in starting positions than boys. The lack of a sex difference for the starting positions shows that this was not the case in our study. Importantly, the developmental trajectory of spatial strategies using egocentric or allocentric cues differs across the lifespan [27,28], possibly affecting sex differences observed at specific points throughout childhood.

It should be noted that an allocentric strategy could also be applied successfully in the identical starting position condition using the relations between available cues irrespective of one’s own position. Contrarily, an egocentric strategy could only be applied successfully in the alternative starting position condition by first navigating back to the starting position of the encoding phase, after which the original path could be re-walked. However, if it were the case that girls were more prone than boys to using the egocentric strategy in the different starting positions condition, we would have observed a sex difference in circumnavigation between the two starting position conditions, which we did not. To enable direct comparison of different spatial strategies between boys and girls, it is crucial to test specifically the degree to which players rely on proximal and distal cues. This could be achieved by optimizing our design by introducing conditions in which either the proximal or distal cues are omitted, and investigating the degree to which these conditions affect performance in boys and girls and over age.

The current study investigated children’s behavioural patterns during navigation and object-relocation. The study design, including the naturalistic virtual reality-based spatial task, provides the opportunity for future studies to incorporate functional magnetic imaging (fMRI) to investigate children’s neural responses. Previous fMRI studies observed distinct brain regions underlying allocentric and egocentric spatial representations [45]. Regions in the temporal lobe—in particular the hippocampus, including the parahippocampal gyrus—are important for allocentric spatial representations (see [46] for a critical review on the neural correlates of allocentric spatial representations), whereas egocentric representations activate the frontal gyrus and parietal regions [47]. Most of this research is focused on adults, and research in children is more sparse. Van Ekert and colleagues (2015) investigated the neural correlates for different landmark types in children aged 8–18 [48]. They observed age-related increases in the parahippocampal region and the anterior cingulate cortex for landmarks that were associated with a relevant spatial context as compared to ambiguous landmarks. Another study, by Murias et al. (2019), compared children’s orientation and navigation abilities with those of adults, and observed distinct neural networks, showing increased connectivity from the right (para)hippocampal gyrus to the caudate nucleus, the insular cortex, and the posterior supramarginal gyrus for adults compared with children. Contrarily, children showed increased connectivity from the right paracentral lobule to the right superior frontal gyrus, compared with adults [49]. Distinct neural correlates for girls and boys were not observed, which could be due to the low sample size, as indicated by the authors. Giedd et al. (1999) investigated structural changes in grey matter in children and adolescents comparing boys and girls, and found that girls show an earlier development of grey matter volume in the frontal and parietal lobe than boys, whereas boys show a slightly earlier development of temporal grey matter [50]. How these structural changes in boys and girls might relate to differences in functional activity requires further investigation.

One limitation of our study is its exploratory nature and limited sample size. Therefore, the current findings should be interpreted with caution, especially when interpreting null effects, as we may have been underpowered to detect more subtle differences, for example between girls and boys. However, our findings can form the basis for future confirmatory studies using a larger sample size to further investigate the results observed here. With respect to the generalizability of our findings based on the characteristics of our sample, we acknowledge that the children in our sample were of relatively high socioeconomic status, based on the educational level of the parents. We corrected for parental education level in our analyses. Nevertheless, results might be different in a sample with different characteristics, especially as socioeconomic status has been found to modify the sex difference in spatial skills [51]. Future research in this area could make an effort to include a wider range of socioeconomic backgrounds to investigate the degree to which our findings are applicable to the population. Furthermore, we included only Dutch children in our study, whereas the sex difference in spatial skills has been found to vary cross-culturally [52]. The generalizability of our findings on a virtual spatial navigation task may also be of concern when evaluating to what degree our findings are applicable to real life navigation. However, performance measures between virtual and real navigation are typically correlated [53], and numerous studies have demonstrated virtual reality tasks to be suitable for testing spatial skills in various populations [24,54,55,56,57], including children [25,34,35,41]. Moreover, we included several characteristics (e.g., 3D environment, first-person perspective, and a variety of naturally available cues) to the task to closely match naturalistic navigation. In order to answer remaining questions regarding age and specific developmental trajectories across age in relation to spatial cognitive development, it would be useful for future studies to use the paradigm presented here to investigate performance of adults and older children for comparison. Ideally, future studies could consider a wider age range, and adopt a longitudinal study design to assess developmental changes in spatial cognition, taking into account sex differences. Such an approach could help pinpoint at what age sex differences start to emerge, and elucidate how these develop over time into adulthood.

Our study adds to the ongoing discussion on the factors contributing to the observed sex differences in spatial skills. This discussion is particularly relevant given the importance of spatial skills for the science, engineering, and mathematics (STEM) domain for the current information age. Modern society requires more people with the right skill set in this particular domain to prevail, and girls remain underrepresented [58]. Identifying the factors contributing to sex differences in spatial cognition could aid in shaping training programmes to help children in general, and girls specifically, to develop the full potential of their abilities. Research in this area could be particularly fruitful given the malleability and transferability of spatial skills [1,5,6,7]. Optimal training could help amplify and diversify STEM achievement, which in turn benefits modern society as a whole.

## 5. Conclusions

In this study, we used a naturalistic virtual reality-based task to demonstrate that sex differences in spatial skills, including navigation and object-relocation, favouring boys are present in 9–11-year-old children. Additionally, the change in performance with age is different for boys and girls in this age range. We also showed that, at this age, boys already have more gaming experience than girls, and that gaming experience had no effect on any of the outcome measures in our study. We encourage future confirmatory studies to test the findings observed in our study, and to apply their findings onto the development of specific trainings for girls and boys to enhance spatial skills.

## Figures and Tables

**Figure 1 brainsci-11-00886-f001:**
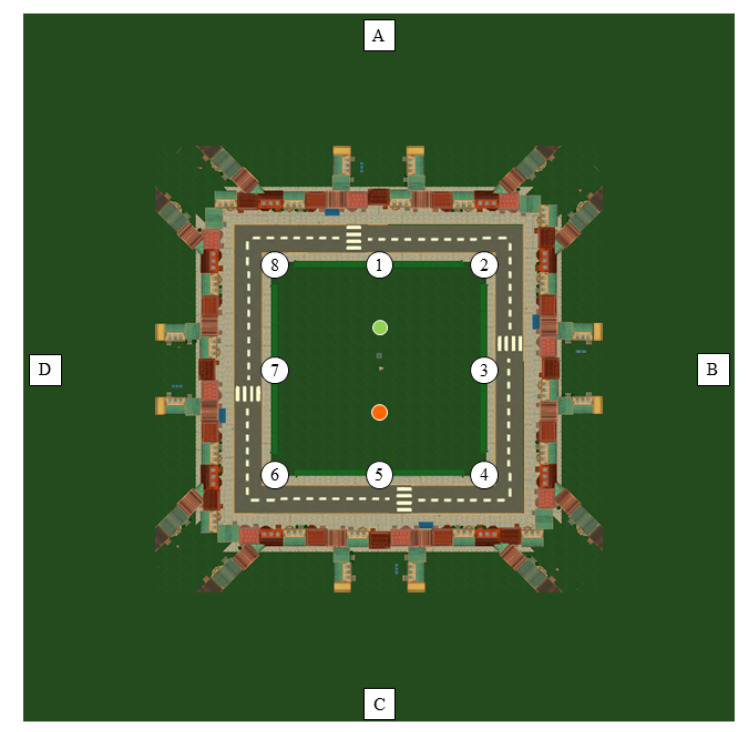
Schematic top view of the virtual environment. White circles with numbers 1–8 represent the possible starting positions (i.e., entrances to the park). The bright green and orange dot in the middle represent two trees with distinctly coloured leaves that served as proximal cues. White squares with letters (**A**–**D**) represent four distinct large buildings that served as distal cues. Note that the distance of the distal cues to the park has been scaled down, and that streets further away from the park have been left out of the image, for illustrative purposes.

**Figure 2 brainsci-11-00886-f002:**
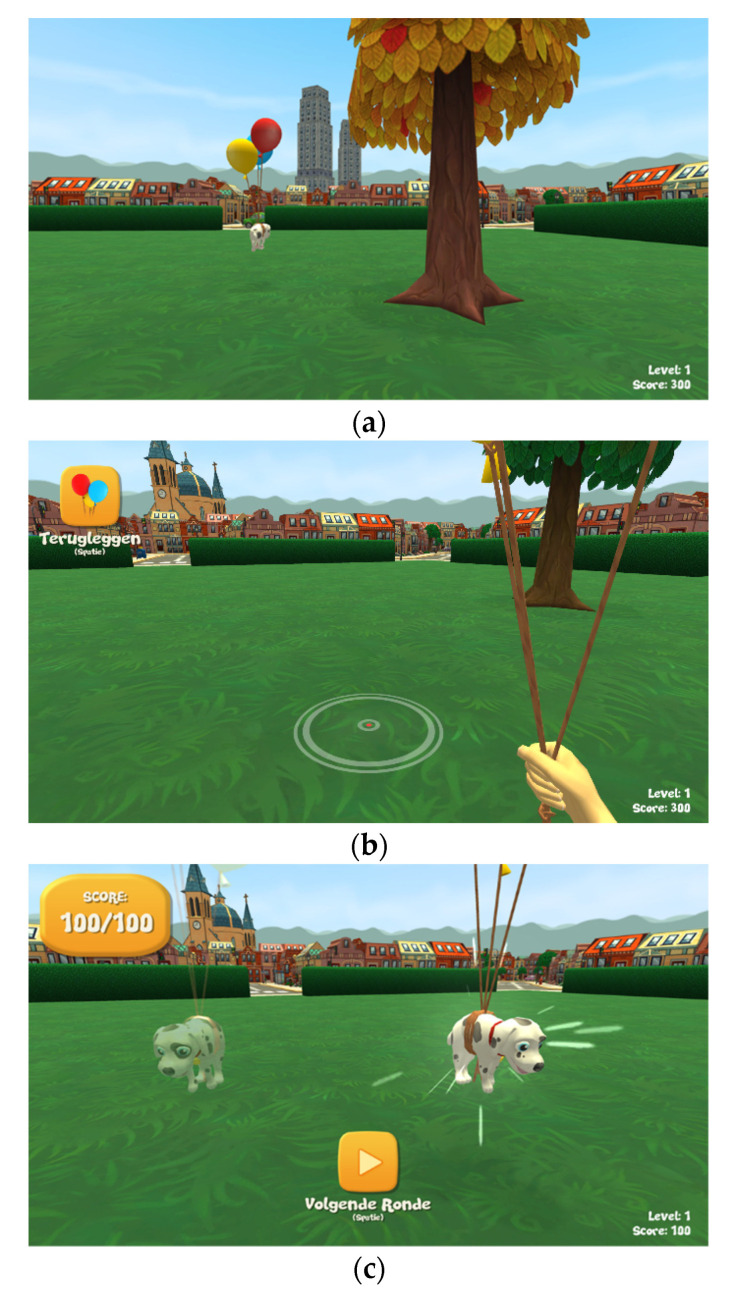
Example screenshots of the player’s view in the different phases of a trial in the spatial cognition game. (**a**) Encoding phase. The object is presented in the middle. Balloons were added to increase visibility from afar. One of the proximal cues (tree with orange leaves) is visible in the front, as well as one of the distal cues (skyscrapers) in the back. (**b**) Testing phase. Player is holding the object by the balloons. The balloon icon on the upper left indicates that the player should now replace the object. The grey circles on the ground and red dot in the middle indicate the exact position where the object is replaced if the space bar is pressed. One of the proximal cues (tree with green leaves) is visible near the player, and one of the distal cues (church) is visible in the back on the left side of the screen. (**c**) Feedback phase. Obtained points are presented on the upper left. The chosen location by the player is show in transparent, and the original location of the object is shown in solid, with surrounding flashing lights.

**Figure 3 brainsci-11-00886-f003:**
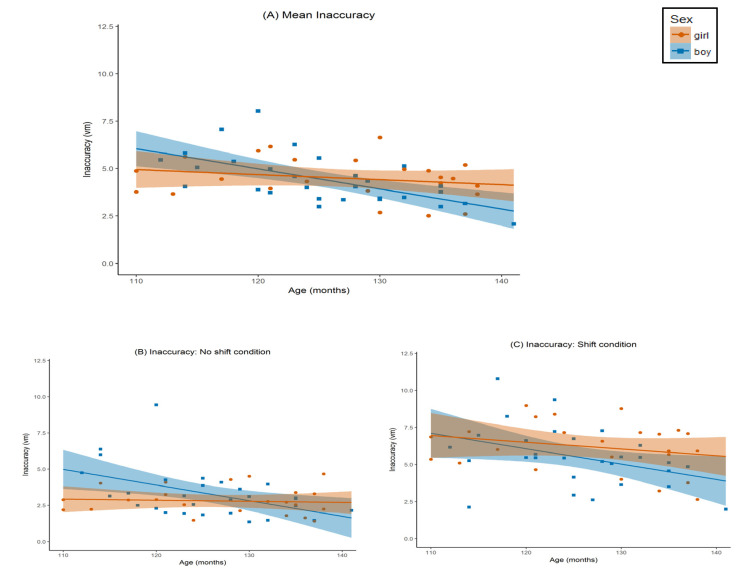
Object distance by age for boys and girls. Object distance was measured as the object distance in virtual meters (vm) between the original location of the object in the encoding phase, and the chosen location by the player in the testing phase. Age was measured in months. The graph is separated by sex: orange circles represent girls, blue squares represent boys. Lines represent the regression line, separated by sex (orange for girls, blue for boys). Shaded area around the regression line represents standard error. (**A**) Mean object distance over both trial types (‘no shift’ and ‘shift’). (**B**) Object distance for the ‘no shift’ condition. (**C**) Object distance for the ‘shift’ condition.

**Figure 4 brainsci-11-00886-f004:**
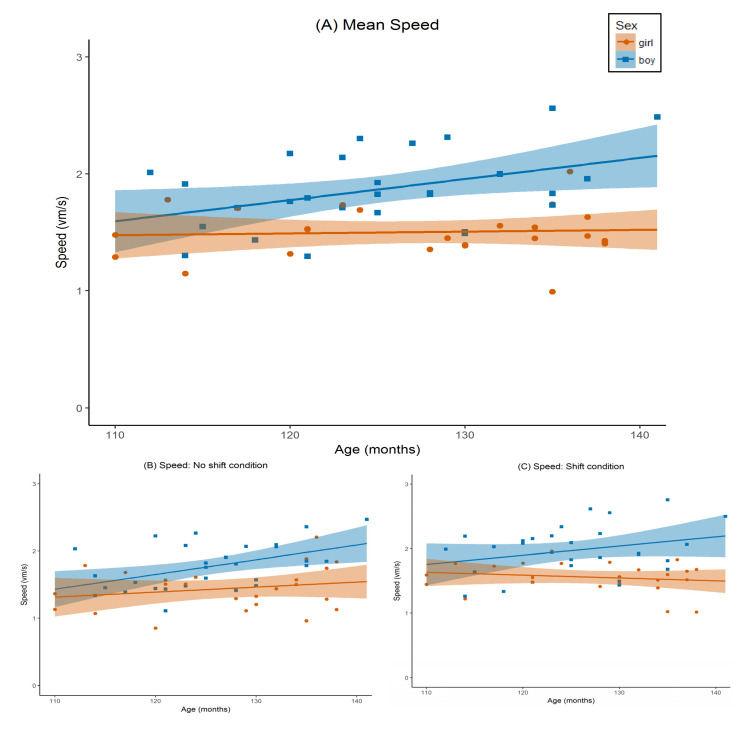
Speed by age for boys and girls. Speed represents average navigational speed of the player in the testing phase in virtual meters per second (vm/s). Age was measured in months. The graph is separated by sex: orange circles represent girls, blue squares represent boys. Lines represent the regression line, separated by sex (orange for girls, blue for boys). Shaded area around the regression line represents standard error. (**A**) Mean speed over both trial types (’no shift’ and ‘shift’). (**B**) Speed for the ’no shift’ condition. (**C**) Speed for the ‘shift’ condition.

**Figure 5 brainsci-11-00886-f005:**
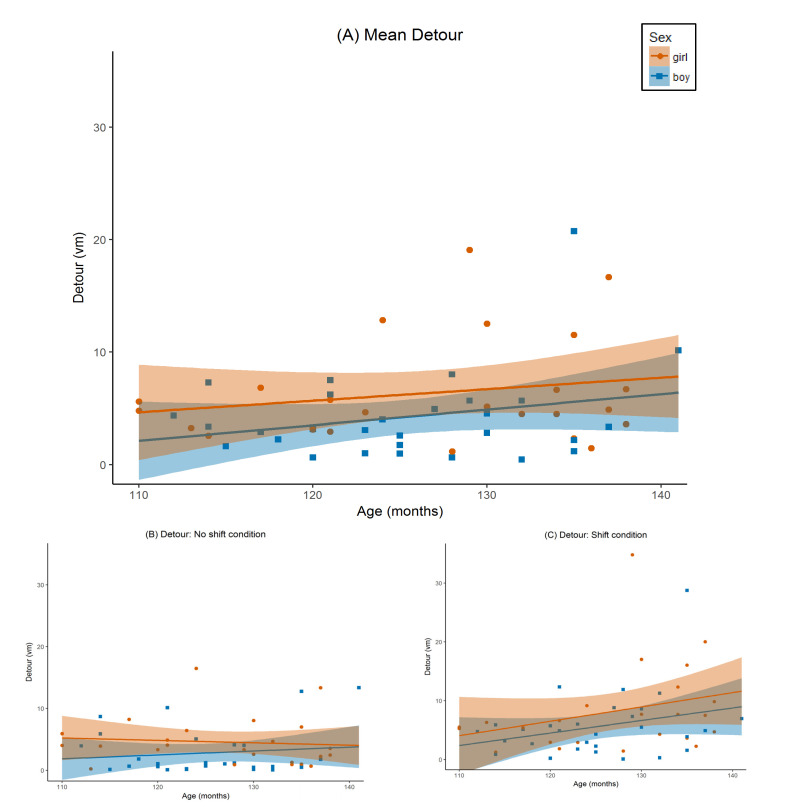
Detour by age for boys and girls. Detour was measured as the walked virtual meters (vm) by the player in the testing phase minus the length in vm of the shortest path between the start and end position of the player in the testing phase. Age was measured in months. The graph is separated by sex: orange circles represent girls, blue squares represent boys. Lines represent the regression line, separated by sex (orange for girls, blue for boys). The shaded area around the regression line represents the standard error. (**A**) Mean Detour over both trial types (‘no shift’ and ‘shift’). (**B**) Detour for the ’no shift’ condition. (**C**) Detour for the ‘shift’ condition.

**Table 1 brainsci-11-00886-t001:** Demographic statistics, separated by sex.

Variable	Sex						
	Girls (*n* = 24)		Boys (*n* = 29)				
	*M*	*SD*	*M*	*SD*	*T*	*p*	Cohen’s *d*
Age in Months	126.92	9.29	125.38	7.55	0.67	0.51	0.18
Gaming Experience	−0.38	0.88	0.32	0.69	−3.24	0.002	0.89
Parental Education	3.50	0.78	4.17	0.66	−3.40	0.001	0.93

**Table 2 brainsci-11-00886-t002:** Means, standard deviations, and correlations of the dependent variables of the repeated measures-MANCOVA.

Variable	*M*	*SD*	1.	2.	3.
1. Object distance	4.45	1.21			
2. Speed	1.70	0.34	*r* = −0.18, *p* = 0.19		
3. Detour	5.20	4.44	*r* = −0.36, *p* = 0.009 **	*r* = 0.03, *p* = 0.82	

** *p* < 0.01.

## Data Availability

The data presented in this study are available on request from the corresponding author. The data are not publicly available because no written consent for public data sharing was obtained from the parents of the participating children.

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
