# Peer review of "Sex Differences and the Role of Gaming Experience in Spatial Cognition Performance in Primary School Children: An Exploratory Study"

_brainsci, 2021, doi:10.3390/brainsci11070886_

Round 1

Reviewer 1 Report

The authors have responded to address the reviewers’ concerns; however, I am in agreeance with Reviewer 3 that the results and their broader contribution to the literature is incremental. This appears to be a data set that was collected for another purpose, likely neuroimaging based on their responses, yet the proper development of this dataset has not been done to round out this “exploratory” study into decipherable and meaningful findings.

Further data is necessary to explore sex differences between children age 9-11 years in this paradigm. Group means (females: N = 24; males N= 29) are not large enough to add to the overall contributions of sex differences to the development of spatial cognition. The participant group is too narrow. A power analysis would be required to demonstrate that the current data is able to explain behavioral variance within the sample. If this is an introduction to a novel paradigm intended for wider use within a developmental sample than expanded sampling is necessary to understand performance using this task.

Furthermore, the task has not been tested in older children and adult age ranges. While similar object-location tasks have been used in adult samples to study spatial cognition quite a bit (referencing Doeller et al., 2008; 2010; Jacobs et al., 2013), the paradigm itself has been modified here quite substantially for use in a developmental sample. Hence, making a comparison of developmental performance to the neural mechanisms contributing to global understanding of our positioning in the world (as typically tested with the object-location task) is quite difficult without having a developed sample to compare against.

To initial points raised, the title of this exploratory analysis references sex differences and the contribution of video game experience to the development of spatial cognition. Within the manuscript’s samples, there is shown no relationship between VG experience, likely for reasons previously mentioned, and sex and the interaction between sex and navigational performance is only found in that males reach their object location faster, yet are not more accurate than females. To explore the contribution of faster navigation times within the males, correlation between RT and accuracy would be necessary at the trial level. Do trials in which males navigate to the object location lead to more accurate responses?

While this introduction to a novel task modified for a developmental sample falls short, the potential for further studies using this task is of high interest.

Reviewer 2 Report

I can see where the authors are coming from in their response to my one major concern with this manuscript. The revision is very thorough in making it clear that this is intended as an exploratory study of the method and theory. It is appropriately flagged in the title and abstract. I have no further concerns that are pressing enough to delay publication. 

Author Response

We are pleased to hear that we have addressed Reviewer 2’s initial concerns to their satisfaction, and that they have no further concerns pressing enough to delay publication.

Round 2

Reviewer 1 Report

I appreciate the added discussion points and analyses to address the reach of the current data. By adding open questions, cautionary statements about interpretations, as well as points of future interest, I am satisfied with the authors' response to my comments and their responses to the other reviewers. 

This manuscript is a resubmission of an earlier submission. The following is a list of the peer review reports and author responses from that submission.

Round 1

Reviewer 1 Report

This manuscript describes the spatial abilities of 9-11 year-old boys and girls in a virtual reality-based task taking into account the gaming experience. Authors showed that boys were faster than girls and differences did not depend on their gaming experience.

Overall, the topic is interesting, the manuscript is well written and it is methodologically correct. Nevertheless, there are few points that need attention:

  • Lines 129-130. Participants: Connect participants’ description to table 1 (demographic info) .
  • Lines 172-177. Did the groups tested on Donders Institute and their own primary school differed in composition (boys-girls)? Please add this information to the manuscript.
  • Figure 2. Images should be ordered and marked with letters a, b or c, according to the legend.
  • How was the distance/score relationship? Please add information about the scores according to the error of estimation.
  • How long it took the assessment in boys and girls? Did any participant exceed 45 minutes in the second level?
  • How many trials completed boys and girls in each level? This information could be compared considering the training level. It could inform about learning differences in both groups.
  • Discussion: It would be appropriate including information regarding the neural base of these differences. How is developing the brain in girls and boys to produce such behavioral differences? Which brain networks/structures could support these patterns of behavior?

Reviewer 2 Report

The authors examined the importance of environmental factors such as gaming experience on sex differences observed in spatial cognition tasks. Sex differences in spatial cognition are evident as the authors note in their Introduction, yet the specific nature of such differences is not well understood. As mentioned here, under varying circumstances tested in spatial tasks, sex differences related to behavior may or may not appear. Hence, this article speaks to a timely topic by using a novel paradigm that would allow for the examination of development of spatial cognition from multiple facets. The paradigm is a highlight of the study. An object-location task has been used often in adult studies of spatial cognition, and development of an adapted task for use in children is highly intriguing. Future results of spatial developmental trajectories in such a design will have a high impact.

However, the exploratory analysis fails to capture the essence of the question proposed. In terms of sex difference, male and female children were equal in accuracy in terms of their learning precision. Yet, males reached the goal at a faster rate. While navigation rate (RT) shows a sex difference, it is a less compelling measurement of overall understanding and learning within the task. While the authors find no overall difference in spatial accuracy across sexes they may further explore learning across trials (does accuracy on trial n+1 show an improvement from trial n), examination of biases toward proximal landmarks, as the authors suggest is likely, angular error, angle of initial movement (as trials progress does their first movement indicate an understanding of where they want to go i.e. the goal location), etc. There is a rich dataset potential here and perhaps these are interrogations the authors are considering.

Then, in terms of measuring video game experience, the authors may not be capturing the essential features of video game play that have been shown to relate to differences in spatial orientation and navigation. Likely only asking the hours/week of video game usage is not capturing the variance that would explain sex differences in behavior. Nora Newcombe and Giuseppe Iaria have more expansive questionnaires and have shown that it is important to ask what video game people play (visual perspective, active or passive component, etc.). It is an important factor in measuring the effects of gaming on spatial orientation and navigation. 

Reviewer 3 Report

This study is looking at the role of sex differences and gaming experience as a predictor for performance on a kind of navigation and location memory task. They report little of either – just that boys performed more quickly, but not more accurately.

The core idea for the study is sound, if a little incremental. The method also seems fine. The analysis makes sense and the results are presented well. The discussion largely makes sense – though it does end the article with the sentence “This section is not mandatory but can be added to the manuscript if the discussion is unusually long or complex.”

The Achilles heel here, really, is just sample size. I am sorry but I cannot give a positive review to a study looking for sex differences in spatial cognition with only 53 participants and no explicit power analysis. I do developmental studies of spatial cognition myself and I am keenly aware of how difficult it can be to collect large samples, so I have some sympathy for the authors. But if you cannot get large samples, you shouldn’t be studying sex differences. Even the effects that have been replicated enough to be convincing can be very small. This is compounded by the specifics of the study as well. Previous research, cited here, looked at a sample of N = 180 to examine differences on a mental rotation task. In the task used here, I would tend to expect even smaller effects and thus expect an even larger sample. The general question of sample size is also compounded by the fact that the authors want to examine two correlated predictors: sex and gaming experience. All of this makes me strongly suspect that the study is seriously underpowered.

I would strongly encourage the authors to use previous research to perform an explicit power analysis. Maybe I’m wrong about the sample size needed. If not, maybe the authors have some ability to collect more data? I could see this project adding usefully to the literature if the sample size was tightly justified and up to par.